

# Subject specific muscle synergies and mechanical output during cycling with arms or legs

Théo Cartier[1], Laurent Vigouroux[1], Elke Viehweger[2,3] and Guillaume Rao[1]

[1] Aix Marseille Univ, CNRS, ISM, Marseille, France
[2] Department of Biomedical Engineering, University of Basel, Allschwil, Switzerland
[3] Department of Orthopedics and Gait Laboratory, University Children's Hospital of Both Basel, Basel, Switzerland

## ABSTRACT

**Background**. Upper (UL) and lower limb (LL) cycling is extensively used for several applications, especially for rehabilitation for which neuromuscular interactions between UL and LL have been shown. Nevertheless, the knowledge on the muscular coordination modality for UL is poorly investigated and it is still not known whether those mechanisms are similar or different to those of LL. The aim of this study was thus to put in evidence common coordination mechanism between UL and LL during cycling by investigating the mechanical output and the underlying muscle coordination using synergy analysis.

**Methods**. Twenty-five revolutions were analyzed for six non-experts' participants during sub-maximal cycling with UL or LL. Crank torque and muscle activity of eleven muscles UL or LL were recorded. Muscle synergies were extracted using nonnegative matrix factorization (NNMF) and group- and subject-specific analysis were conducted.

**Results**. Four synergies were extracted for both UL and LL. UL muscle coordination was organized around several mechanical functions (pushing, downing, and pulling) with a proportion of propulsive torque almost 80% of the total revolution while LL muscle coordination was organized around a main function (pushing) during the first half of the cycling revolution. LL muscle coordination was robust between participants while UL presented higher interindividual variability.

**Discussion**. We showed that a same principle of muscle coordination exists for UL during cycling but with more complex mechanical implications. This study also brings further results suggesting each individual has unique muscle signature.

Corresponding author
Théo Cartier,
theo.cartier@univ-amu.fr

## INTRODUCTION

Cycling with the upper (UL) or lower limbs (LL) both consists in producing torque in a circular movement with fixed cranks length around a fixed rotation axis. Cycling is extensively used for improving health, training, physiological evaluation, or rehabilitation with LL (*Faria, Parker & Faria, 2005*; *Fujiwara PhD et al., 2005*; *Oja et al., 2011*; *Sitko et al., 2020*) and to a lesser extent with UL (*Hübner-Wozniak et al., 2004*;

*Elmer, Danvind & Holmberg, 2013*; *Zinner et al., 2016*; *Zhou et al., 2017*). An increasing number of studies highlight the interest of UL cycling also for LL rehabilitation: for example, *Kaupp et al. (2018)* showed functional and neurological improvements of walking capacities after a rehabilitation protocol involving UL cycling. Moreover, modulations of cutaneous reflex and muscle activities of the LL was also shown after UL cycling training (*Zehr & Chua, 2000*; *Zehr, 2005*; *Barzi & Zehr, 2008*; *Mezzarane, Nakajima & Zehr, 2014*). These examples suggest similarities in movement execution and interactions in the production of movement with the UL and LL that can be observed at muscle activity level. This latter point is particularly interesting from a clinical point of view since intervention at the UL could benefit the LL functional aspects.

To deeply understand the movement of cycling, it remains crucial to understand the underlying muscle coordination. Muscle coordination leads to the question of understanding how the large number of available muscles are coordinated to execute a movement (*Bernstein, 1967*; *Latash, 2012*). The muscle synergy analysis is frequently used to answer this question (*d'Avella, Saltiel & Bizzi, 2003*; *Turpin, Uriac & Dalleau, 2021*). This analysis assumes that the Central Nervous System (CNS) simplifies the abundance of muscles by using smaller numbers of functional elements activated in time. Muscles synergies thus allow a global understanding of the muscle coordination in comparison to the production of a mechanical output. The existence of muscle synergies on the LL during cycling has been widely demonstrated even when manipulating various mechanical constraints such as posture or output torque intensity (*Hug et al., 2011*). During LL cycling, it has been shown that muscle synergies are shared across participant despite interindividual variability of EMG pattern (*Hug et al., 2010*; *De Marchis et al., 2013*). Further studies showed that this movement is executed, from a kinematic and kinetics point of view, in the same way by any healthy person, expert or not (*Mornieux et al., 2008*).

In comparison to LL cycling, UL cycling muscular coordination remains poorly understood despite the interests for training and clinical applications. A single recent study aimed to extract muscle synergies during UL cycling (*Botzheim et al., 2021*). Using a set of four muscles for each arm, these authors showed that bimanual muscle coordination can be described by muscle synergies. However, no simultaneous analysis of the mechanical output was performed and only a few numbers of representative muscles were used to fully understand the functional synergies during UL cycling. More generally, it remains unknown whether the UL synergies are similarly organized than in LL with respect to the mechanical output. Unfortunately, this lack of knowledge limits the potential for these exercises to be objectively used for training and clinical applications.

Complementary, several studies highlight the necessity to take into account individual differences (*Bartlett, Wheat & Robins, 2007*) and a recent work of *Hug et al. (2019)* put in evidence that individuals have unique muscle signatures. Each individual muscle signature could result from numerous individual factors such as development, experiences, or diseases discrepancies. Contrary to LL, UL musculoskeletal system offers the possibility to pull or push equally, allowing several possible strategies in movement execution. Individual differences in synergies can thus be differently managed in the UL compared to LL.
The first objective of this study was thus to fill the knowledge gap by investigating the simultaneous temporal evolutions of the mechanical output and of the EMG and muscle synergies of the Upper Limb in comparison to the Lower Limb during cycling. To individualize our analysis, our second objective was to compare interindividual variability between UL and LL at kinematics, kinetics, and muscle coordination levels.

# MATERIALS & METHODS

## Participants

Six young healthy adults (age: $27.2 \pm 2.9$ years; weight: $69 \pm 7.5$ kg; height: $174.2 \pm 5.1$ cm) volunteered to participate in the study and suffered of no injuries in the past 6 months. To limit the effect of participants' level, expertise in either UL or LL cyclic exercise constituted an exclusion criterion. Participants gave their written consent and were informed of the possible risks and discomfort associated with the experiment. The experimental protocol was approved by the national ethic committee (CERSTAPS, IRB00012476-2020-30-11-75) and the study was conducted in accordance with the Helsinki declaration.

## Experimental procedure

The experimental session consisted in two phases performed successively with the UL or the LL in random order. Before each phase, participants performed a 5-minutes standardized warm up with the tested limb at self-selected cadence (SSC: $85.0 \pm 10.4$ *vs* $86.3 \pm 9.8$ rpm, $p > 0.05$, for arms and legs respectively). The self-selected cadence was then used and controlled for the entire protocol. To estimate maximal power (MP), participants performed a single 10 s isokinetic all-out with strong verbal encouragements. MP was computed as the mean peak power during the 10 s. The same relative amount of power was set at 30% of MP to reach a significant intensity without any excessive fatigue and to compare both limbs at the same relative mechanical demands. Participants then performed a submaximal 2-minute test at 30% of MP at self-selected cadence ($102.8 \pm 14.7$ *vs* $171.7 \pm 34.4$ W, $p < 0.05$, for arms and legs respectively). The two experimental phases were separated by at least 35 min rests to limit the fatigue effect and change the experimental equipment (EMG, markers *etc.*) from one limb to another.

## Material and data collection
### Ergometer

The task was performed on a cycle ergometer (Excalibur for LL and Brachumera for UL, Lode, Groningen, The Netherlands) equipped with standard cranks, flat pedals for feet and standard horizontal pegs for hands. The type of pedals interface was chosen to study the most ecological situation for each limb. For LL cycling, participants were asked to seat comfortably on the saddle and maintain their position during the entire experimental procedure. For UL cycling, participants were seated in front of the ergometer in pronated grip (Fig. 1). The crank axis height was positioned at the glenohumeral level. Participants were preferentially seated on a chair and were instructed not to grab the chair legs with their own legs. Torque and pedalling frequency were recorded each 2° by the Lode Ergometry Manager software. Ergometers were calibrated before each evaluation.

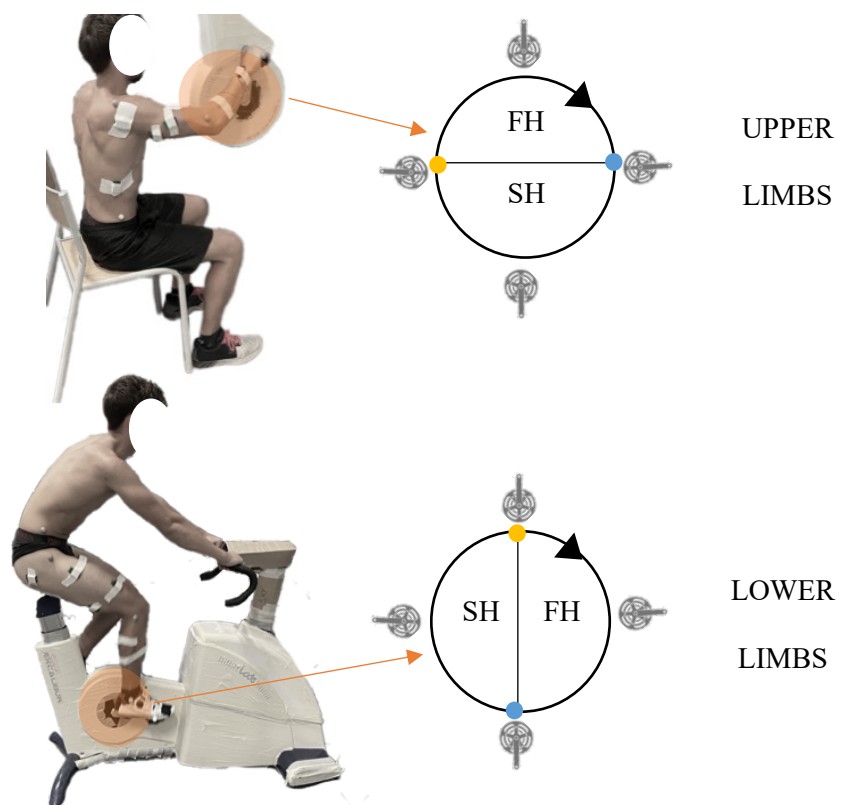

**Figure 1** **Experimental set-up.** Left column: Example of experimental set-up for UL (top) and LL (bottom) and schematic representation of the cycle revolution for each. The yellow dot represent the SH to FH transition and the blue dot the FH to SH transition. FH, first half, SH, second half.

### Kinematics

Kinematics was recorded using 5 Qualisys cameras (Oqus 7) at 200 Hz with Qualisys Track Manager software (Qualisys, Sweden). Trajectories of five reflective markers placed on distal extremity of the 5[th] *metacarpal, ulnar styloid, lateral epicondyle, acromion* and *iliac crest* and distal extremity of the 5[th] *metatarsal, lateral malleolus, lateral knee epicondyle, greater trochanter* and *iliac crest* were used to compute the 2D angular trajectories of the ankle, knee, hip and wrist, elbow, shoulder joints.

### EMG

The electromyography of 22 muscles were recorded using EMG system (Wireless System, DELSYS, USA) sampled at 2000 Hz with the Delsys Trigno Acquisition software. Before electrode application, the skin was shaved and cleaned with an alcoholic solution to improve the electrode/skin impedance. SENIAM recommendations were followed to ensure correct electrode placement and avoid crosstalk between EMG signals. A functional test was carried out for each muscle to ensure correct electrode placement. The following LL muscles were recorded: *Tibialis Anterior* (TA), *Soleus* (Sol), *Gastrocnemius Medialis/Lateralis* (GM, GL), *Vastus Medialis/Lateralis* (VM, VL), *Biceps Femoris* (BF), *Semimembranosus* (SM),

*Rectus Femoris* (RF), Tensor Fasciae Latae (TF) and *Gluteus Maximum* (GMAX) (*Hug & Dorel, 2009*; *Hug et al., 2010*; *Hug et al., 2011*; *De Marchis et al., 2013*). The following UL muscles were recorded: *Extensor Carpi Radialis Longus* (ECRL), *Flexor Carpi Radialis* (FCR), *BrachioRadialis* (BR), *Biceps Brachii* (BB), *Triceps Brachis Longus/Shortus* (TBL, TBS), *Deltoid Anterior/Posterior* (DA, DP); (*Zehr & Chua, 2000*; *Barzi & Zehr, 2008*; *Zehr, Loadman & Hundza, 2012*; *Chaytor et al., 2020*), *Pectoralis Major* (PM), *Latissimus Dorsi* (LD), *Flexors Digitorum Superficialis* (FDS). EMG, torque, and kinematic recording were synchronized via a trigger impulse.

## Data processing

### Torque and kinematics

Marker trajectories and torque data were low pass filtered (4th order, zero-time lag Butterworth 6 Hz). Absolute 2D angle of the LL (Hip, Knee, Ankle) and UL (Shoulder, Elbow Wrist) were computed using vector coordinates of each segment in a sagittal plane of the ergometer. Pedalling revolutions were extracted using the position of the distal marker. For LL, 0° was defined as the higher position of the pedal and 180° as the lower position. For UL, 0° was define as the nearest position of the pedal, when the crank is horizontally directed towards the participant and far 180° the farthest when the crank is horizontally directed to the opposite of the participant. Two phases are considered in the movement, the First Half (FH) from 0° to 180° and the Second Half (SH) from 180° to 360°. The FH to SH transition was thus when the limb is fully extended and the SH to FH transition when the limb is grouped. To normalize torque between participants, absolute values were divided by body mass.

### EMG pre-processing

EMG data were first filtered using a bandpass 4th order Butterworth zero-time lag filter (20–400 Hz) then full wave rectified and low pass filtered (4th order Butterworth, zero-time lag 4 Hz) to obtain EMG envelops. For amplitude analysis, the EMG data were normalized by the maximal value recorded during the 10 s all-out. A dedicated detection algorithm has been used to select the 25 most representative cycles over the entire dataset (*Sangeux & Polak, 2015*).

After revolution extraction and pre-processing, torque, kinematics, and EMG data were resampled to 200 points by revolution.

### Synergy extraction

Since the aim of our study was to quantify muscle coordination, EMG data were normalized by the maximal value across the cycles before synergy extraction and synchronous muscle synergy model was used to extracted synergies Eq. (1):

$$V_O(t) = V_R(t) + \varepsilon(t) \text{ with } V_R(t) = \sum_{s=1}^{K} H_s(t) W_s \tag{1}$$

where $V_O(t)$ indicates the initial EMG values of all muscles at time instant t, $V_R(t)$ is the reconstructed signal resulting from the linear combination of $W_s$ (the muscle synergy vector for the s-th synergy) and $H_s(t)$ (the value of the synergy activation coefficient for
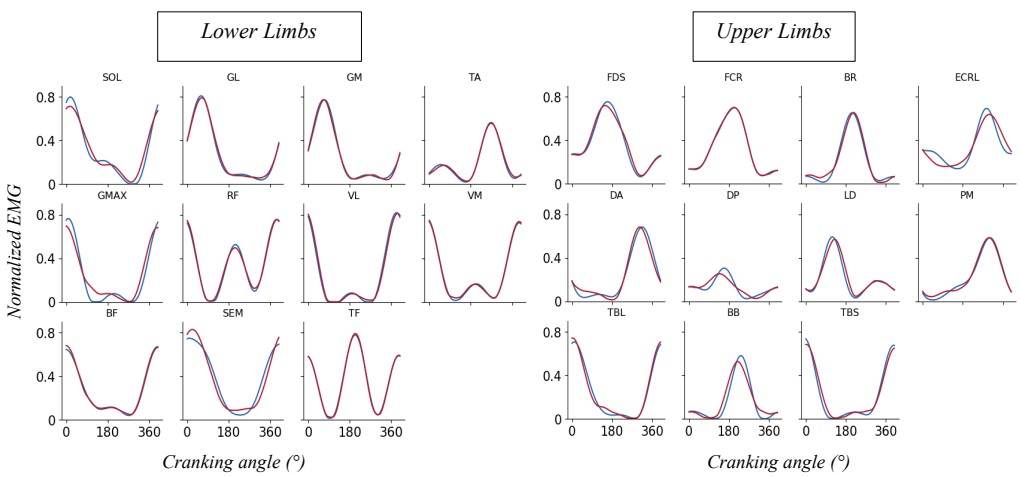

**Figure 2  Reconstructed (red) *vs* original (blue) EMG pattern.** Example of reconstructed *vs* original EMG pattern for representative participant for lower limb muscles (left) and upper limb muscles (right).

the s-th synergy at time instant t). $K$ is the number of synergies necessary to reconstruct the initial EMG dataset, and $\varepsilon(t)$ represents noise. This model is a compromise that allows the analysis of both spatial (through W) and temporal (through H) structures of the motor outputs (*Turpin, Uriac & Dalleau, 2021*).

Nonnegative Matrix Factorization (NNMF) algorithm was applied to a matrix containing the envelop of the 11 muscles across 25 cycles. The algorithm approximates the initial signal $V_O$ with a reconstructed matrix $V_R = WxH$ by minimizing the Frobenius norm $\|V_O - V_R\|$ where $W$ is a $K \times 11$ matrix of the synergy vectors and $H$ is a $K \times 5000$ (25 cycles * 200 time points). The algorithm was run for K in range from 1 to 10 and 10 times for each K synergies combinations to avoid local minima and keep the best solution for further analysis.

The number of synergy sufficient to reconstruct the original matrix $V_O$ was set by the calculation of the Variance Accounted For (VAF) by $V_R$ on the entire matrix (VAF$_{tot}$) and for each muscle individually (VAF$_m$) with following equations:

$$VAF_{tot} = 1 - \frac{\sum(VO_{tot} - VR_{tot})^2}{\sum VO_{tot}^2} \tag{2}$$

$$VAF_m = 1 - \frac{\sum(VO_m - VR_m)^2}{\sum VO_m^2}. \tag{3}$$

The minimum number of synergies that explain at least 90% of VAF$_{tot}$ and 75% of VAF$_m$ was chosen. An example of reconstructed vs original EMG pattern was shown on Fig. 2. When both VAF constraints are satisfied (*i.e.*, total VAF and by muscle VAF), 4 synergies were extracted for each participant to allow an analysis and a functional comparison between participants.

## Statistical analysis

For each temporal data (Angle, Torque, EMG and Hsyn), the inter- or intra-individual variability was calculated using a Pearson correlation coefficient r ($r_{inter}$ or $r_{intra}$) and a Variance Ratio parameter ($VR_{inter}$ or $VR_{intra}$).

For interindividual variability, the input matrix X was composed of a mean curve across the 25 cycles for each participant (final size = $6 \times 200$). For intraindividual variability, the input matrix X was composed of the 25 cycles for each participant, (final size = $25 \times 200$).

The VR was computed as follow:

$$VR_a = \frac{\sum_{i=1}^{k} \sum_{j=1}^{n} (X_{ij} - \overline{X}_i)^2 / k(n-1)}{\sum_{i=1}^{k} \sum_{j=1}^{n} (X_{ij} - \overline{\overline{X}})^2 / (kn-1)} \text{ with } \overline{\overline{X}} = \frac{1}{k} \sum_{i=1}^{k} \overline{X}_j. \tag{4}$$

where $a$ stand for inter or intra, $k$ is the number of samples ($k = 200$), n is the number of participants/cycles (*i.e.*, 6/25), $X_{ij}$ is the value at the ith sample for the jth participant/cycle, $\overline{X}_i$ is the mean value at the ith sample over the n participants/cycles.

The Variance Ratio parameter has been used in other studies (*Burden, Trew & Baltzopoulos, 2003*; *Hug et al., 2010*; *De Marchis et al., 2013*) as an indicator of the overall variation of the data with respect to a mean profile: the higher the VR, the higher the inter or intra participants variability in the dataset. The Pearson correlation coefficient r was computed for inter individual variability between each pair of participants excluding autocorrelation and a mean ± sd value is presented (Tables 1–2). For intra-individual variability, r was computed between each pair of cycle for each participant excluding autocorrelation and a mean ± sd value across all participants is presented (Tables 3–4). The Pearson correlation coefficient r indicates the similarity in the shape profile of the data without being impacted by amplitude differences. We also used this coefficient r to compute similarities across participant in muscle synergy vectors ($W_s$).

Because VR and r do not take account a possible time shift between two identical curves, the normalized cross-correlation function was computed and allowed to extract the percentage of shift ($t_{max}$) needed to maximize the cross-correlation function ($r_{max}$). $r_{max}$ and $t_{max}$ were only computed for interindividual variability, intraindividual variability (through VR and r) being low for each participant. Even though these four variables (VR, r, $t_{max}$, and $r_{max}$) are redundant, they allow to nuance the interpretation of the variability according to differences in time, amplitude, or patterns. Furthermore, these different data allow an easier comparison of our results with the literature.

We performed correlations between the variables presenting high $t_{max}$ ($H_s$, Torque) and the RPM to further analyze the relationships between the Realization variables and the Performance variables (see Discussion section for the definition). The time-shift of each synergy activation coefficient ($H_s$) was computed by extracting the normalized time instants of max($H_s$) and Pearson r correlations were performed for all participants between each time-shift and pedaling cadence.
**Table 1  Interindividual variability of upper limb variables.**

| | | VR | r (mean ± sd) | $r_{max}$ (mean ± sd) | $t_{max}$ % (mean ± sd) |
|---|---|---|---|---|---|
| KINEMATICS & DYNAMICS | | | | | |
| Torque | | 0.26 | 0.72 ± 0.12 | 0.84 ± 0.10 | 2.37 ± 2.46 |
| Shoulder | | 0.12 | 0.93 ± 0.03 | 0.99 ± 0.01 | 0.87 ± 1.01 |
| Elbow | | 0.27 | 0.84 ± 0.12 | 0.97 ± 0.02 | 0.00 ± 0.00 |
| Wrist | | 0.79 | 0.25 ± 0.26 | 0.81 ± 0.15 | 11.77 ± 9.46 |
| EMG | | | | | |
| FDS | | 0.56 | 0.48 ± 0.42 | 0.81 ± 0.12 | 15.40 ± 12 ± 69 |
| FCR | | 0.45 | 0.57 ± 0.40 | 0.77 ± 0.13 | 8.93 ± 8.93 |
| BR | | 0.73 | 0.28 ± 0.56 | 0.88 ± 0.08 | 16.00 ± 11.41 |
| ECRL | | 0.49 | 0.57 ± 0.33 | 0.91 ± 0.04 | 11.06 ± 7.05 |
| DA | | 0.67 | 0.36 ± 0.49 | 0.86 ± 0.10 | 13.47 ± 10.09 |
| DP | | 1.00 | 0.10 ± 0.50 | 0.71 ± 0.17 | 15.70 ± 10.37 |
| LD | | 0.67 | 0.32 ± 0.49 | 0.87 ± 0.07 | 12.50 ± 8.12 |
| PM | | 0.91 | 0.22 ± 0.40 | 0.73 ± 0.10 | 19.93 ± 14.14 |
| TBL | | 0.46 | 0.53 ± 0.44 | 0.80 ± 0.16 | 23.47 ± 31.18 |
| BB | | 0.61 | 0.38 ± 0.46 | 0.88 ± 0.07 | 13.87 ± 9.15 |
| TBS | | 0.89 | 0.16 ± 0.52 | 0.65 ± 0.12 | 15.37 ± 14.06 |
| Synergies | $W_s$ | | | $H_s$ | |
| $S_0$ | 0.75 ± 0.17 | 0.61 | 0.40 ± 0.48 | 0.80 ± 0.15 | 11.80 ± 9.81 |
| $S_1$ | 0.71 ± 0.11 | 0.41 | 0.46 ± 0.42 | 0.90 ± 0.06 | 12.60 ± 9.70 |
| $S_2$ | 0.75 ± 0.12 | 0.70 | 0.30 ± 0.52 | 0.89 ± 0.08 | 14.57 ± 9.99 |
| $S_3$ | 0.75 ± 0.17 | 0.64 | 0.38 ± 0.46 | 0.88 ± 0.08 | 12.87 ± 9.22 |

**Notes.**
Variance Ratio (VR), r-Pearson (r), maximum value of cross-correlation function (rmax) and time lag (tmax) of Kinematics, torque, EMG patterns, synergies vectors (Ws) and synergy activation coefficients (Hs).

# RESULTS

## Torque

Figure 3 shows the averaged torque applied on the right pedal for UL (top) and LL (bottom). Maximal peak torque was higher for LL than UL ($0.72 \pm 0.12$ Nm/kg *vs* $0.18 \pm 0.30$ Nm/kg). For UL, the output torque was positive during most of the cycle ($85 \pm 9\%$) excepted at the end of SH and maximal torque was produced at the beginning of SH (around $199 \pm 42°$). For LL, the torque was positive during FH ($52 \pm 2\%$ of the total cycle) and negative during SH. Maximal torque was produced about mid-FH ($87 \pm 8°$).

## Joint kinematics

Figure 4 shows the kinematics of proximal, intermediate, and distal joints for LL (left) and UL (right). Hip and knee angular displacements result from extension during FH and flexion during SH. Ankle angular displacement result from plantar flexion in FH then dorsiflexion during SH. Shoulder angular displacement results from antepulsion until mid of FH then retropulsion until almost the end of crank revolution. Elbow angular

**Table 2** Interindividual variaibilty of lower limbs variables.

| | VR | r (mean ± sd) | r$_{max}$ (mean ± sd) | t$_{max}$ % (mean ± sd) |
|---|---|---|---|---|
| KINEMATICS & DYNAMICS | | | | |
| Torque | 0.04 | 0.97 ± 0.01 | 0.99 ± 0.01 | 0.81 ± 0.78 |
| Hip | 0.15 | 0.98 ± 0.01 | 0.99 ± 0.00 | 0.00 ± 0.00 |
| Knee | 0.05 | 0.99 ± 0.00 | 1.00 ± 0.00 | 0.00 ± 0.00 |
| Ankle | 0.50 | 0.76 ± 0.13 | 0.89 ± 0.08 | 2.37 ± 2.77 |
| EMG | | | | |
| SOL | 0.20 | 0.84 ± 0.15 | 0.93 ± 0.04 | 3.40 ± 3.72 |
| GL | 0.37 | 0.68 ± 0.26 | 0.91 ± 0.05 | 9.87 ± 6.10 |
| GM | 0.19 | 0.82 ± 0.17 | 0.97 ± 0.03 | 5.73 ± 3.50 |
| TA | 0.57 | 0.37 ± 0.45 | 0.84 ± 0.10 | 24.93 ± 27.13 |
| GMAX | 0.07 | 0.97 ± 0.02 | 0.97 ± 0.02 | 0.00 ± 0.00 |
| RF | 0.35 | 0.80 ± 0.12 | 0.80 ± 0.12 | 0.43 ± 1.05 |
| VL | 0.05 | 0.96 ± 0.03 | 0.96 ± 0.03 | 0.10 ± 0.39 |
| VM | 0.08 | 0.94 ± 0.05 | 0.94 ± 0.04 | 0.33 ± 0.84 |
| BF | 0.45 | 0.56 ± 0.32 | 0.77 ± 0.14 | 8.90 ± 6.84 |
| SEM | 0.45 | 0.64 ± 0.28 | 0.88 ± 0.06 | 8.70 ± 6.10 |
| TF | 0.75 | 0.25 ± 0.58 | 0.68 ± 0.20 | 25.53 ± 19.91 |
| Synergies | W$_s$ | H$_s$ | | |
| S$_1$ | 0.75 ± 0.13 | 0.06 | 0.95 ± 0.04 | 0.95 ± 0.04 | 0.16 ± 0.52 |
| S$_2$ | 0.73 ± 0.15 | 0.21 | 0.73 ± 0.15 | 0.97 ± 0.02 | 5.73 ± 3.53 |
| S$_3$ | 0.39 ± 0.37 | 0.48 | 0.54 ± 0.36 | 0.83 ± 0.16 | 9.07 ± 5.74 |
| S$_4$ | 0.87 ± 0.11 | 0.30 | 0.75 ± 0.24 | 0.95 ± 0.04 | 6.27 ± 4.90 |

**Notes.**
Variance Ratio (VR), r-Pearson (r), maximum value of cross-correlation function (rmax) and time lag (tmax) of Kinematics, torque, EMG patterns, synergies vectors (Ws) and synergy activation coefficients (Hs).

displacement results from extension during FH then flexion during SH. Wrist angular displacement results from extension during FH then flexion during SH.

## Synergies description
### UL

Four synergies were extracted for each participant (Fig. 5-UL) and explained 96 ± 0.01% of the total variance of the initial EMG dataset (VAF$_{tot}$) and 95 ± 0.05% of each initial individual EMG dataset (VAF$_m$). The **first synergy** (**S$_1$**) included the TBS and TBL during the transition from SH to FH and the first part of FH. The **second synergy** (**S$_2$**) was composed by the activations of the hand flexors (FDS and FCR) and LD during end of FH and the transition from FH to SH (90–180°). The **third synergy** (**S$_3$**) involved hand flexors (FDS/FCR) and extensors (ECRL), elbow flexors (BB and BR), elbow extensors (TBS) and shoulder extensor (DP) during SH (180–270°). The **last synergy** (**S$_4$**) involved DA and PM in anticipation to the SH to FH transition (270–0°).

**Table 3  Upper limbs intra individual variability.**

|  | VR | r (mean ± sd) |
|---|---|---|
| KINEMATICS AND DYNAMICS |  |  |
| Torque | 0.19 | 0.89 ± 0.13 |
| Shoulder | 0.06 | 0.98 ± 0.03 |
| Elbow | 0.15 | 0.94 ± 0.13 |
| Wrist | 0.58 | 0.70 ± 0.33 |
| EMG |  |  |
| FDS | 0.21 | 0.90 ± 0.09 |
| FCR | 0.17 | 0.91 ± 0.08 |
| BR | 0.14 | 0.95 ± 0.07 |
| ECRL | 0.24 | 0.89 ± 0.12 |
| DA | 0.18 | 0.93 ± 0.08 |
| DP | 0.43 | 0.78 ± 0.24 |
| LD | 0.32 | 0.81 ± 0.26 |
| PM | 0.37 | 0.82 ± 0.20 |
| TBL | 0.11 | 0.95 ± 0.05 |
| BB | 0.13 | 0.96 ± 0.06 |
| TBS | 0.21 | 0.87 ± 0.15 |
| SYNERGIES |  |  |
| $H_1$ | 0.11 | 0.95 ± 0.06 |
| $H_2$ | 0.12 | 0.94 ± 0.08 |
| $H_3$ | 0.07 | 0.98 ± 0.02 |
| $H_4$ | 0.13 | 0.95 ± 0.05 |

**Notes.**

Variance Ratio (VR) and r-Pearson (r) of Kinematics, torque, EMG patterns, synergies vectors ($W_s$) and synergy activation coefficients ($H_s$).

### LL

Four synergies were extracted for each participant (Fig. 5-LL) and explained 96 ± 0.02% of $VAF_{tot}$ and 95 ± 0.05% of $VAF_m$. The **first synergy** ($S_1$) involved hip flexors (RF, TF) and extensors (GMAX, BF, SEM), knee extensors (RF, VM, VL) and SOL and was active around the upper transition and the first part of FH (270-90°). The **second synergy** ($S_2$) involved ankle plantar-flexors (SOL, GL, GM), hip extensors (BF, SEM) and occurred during at mid-FH (around 90°). The **third synergy** ($S_3$) located around the bottom transition (180°), involved ankle plantar-flexors (GL) and hip flexors (TF). **The fourth synergy** ($S_4$) involved principally the TA and to a lesser extent the RF and TF during SH (180–0°).

## Interindividual variability

The results concerning the interindividual variability are described in Tables 1 and 2.

### Joint kinematics

UL showed slightly more angular kinematics variability than LL. The distal joint showed more variability than the proximal and intermediate joints for both UL and LL. Negligible time shifts in kinematic patterns were found for both limbs (Tables 1–2 - Kinematics & dynamics).

| Table 4 | Lower limbs intra individual variability. | |
|---|---|---|
| | $VR_{intra}$ | $r_{intra}$ (mean $\pm$ sd) |
| KINEMATICS AND DYNAMICS | | |
| Torque | 0.02 | $1.00 \pm 0.00$ |
| Hip | 0.01 | $1.00 \pm 0.00$ |
| Knee | 0.00 | $1.00 \pm 0.00$ |
| Ankle | 0.14 | $0.96 \pm 0.05$ |
| EMG | | |
| SOL | 0.06 | $0.98 \pm 0.02$ |
| GL | 0.09 | $0.97 \pm 0.04$ |
| GM | 0.07 | $0.82 \pm 0.04$ |
| TA | 0.32 | $0.95 \pm 0.30$ |
| GMAX | 0.14 | $0.90 \pm 0.06$ |
| RF | 0.23 | $0.98 \pm 0.13$ |
| VL | 0.05 | $0.98 \pm 0.03$ |
| VM | 0.06 | $0.97 \pm 0.03$ |
| BF | 0.15 | $0.93 \pm 0.07$ |
| SEM | 0.22 | $0.89 \pm 0.15$ |
| TF | 0.21 | $0.87 \pm 0.18$ |
| SYNERGIES | | |
| $S_1$ | 0.04 | $0.98 \pm 0.02$ |
| $S_2$ | 0.05 | $0.98 \pm 0.02$ |
| $S_3$ | 0.10 | $0.95 \pm 0.05$ |
| $S_4$ | 0.26 | $0.88 \pm 0.19$ |

Notes.
Variance Ratio (VR) and r-Pearson (r) of Kinematics, torque, EMG patterns, synergies vectors ($W_s$) and synergy activation coefficients ($H_s$).

### Torque

UL showed more interindividual variability in torque production than LL ($VR_{UL} = 0.26$, $r_{UL} = 0.72 \pm 12$ *vs* $VR_{LL} = 0.04$, $r_{LL} = 0.97 \pm 0.01$ respectively, $p < 0.05$). A low time shift was found at the maximum of the cross-correlation for both limbs ($t_{max-UL} = 2.37 \pm 2.46$; $t_{max-LL} = 0.81 \pm 0.78$). These results indicates that interindividual variability in torque pattern mainly lies in differences in the pattern that is produced and not in a time shifting of similar patterns. Nevertheless, considering the amount of variability, even with a higher value for UL, a similar pattern of torque is produced by participants (Table 1, Fig. 3).

### EMG

Compared to LL, UL EMG patterns showed high interindividual variability for all muscles ($VR_{EMG-UL} = 0.68 \pm 0.22 > VR_{EMG-LL} = 0.32 \pm 0.22$, $r_{EMG-UL} = 0.36 \pm 0.17 < r_{EMG-LL} = 0.71 \pm 0.24$, $p < 0.05$, Tables 2–3-EMG). For LL, only GL, TA, BF, SEM and TF muscles showed high variability ($VR > 0.37$, $r > 0.64 \pm 0.28$), these muscles being mainly involved in $S_3$ and $S_4$ and not associated to the production of propulsive torque. An important time shift was found at the maximum of cross-correlation function for UL ($t_{max-EMG-UL} = 15,01 \pm 4\%$ of the total cycle, Table 1-EMG) compared to LL ($t_{max-EMG-LL} = 7.9 \pm 9.3\%$,
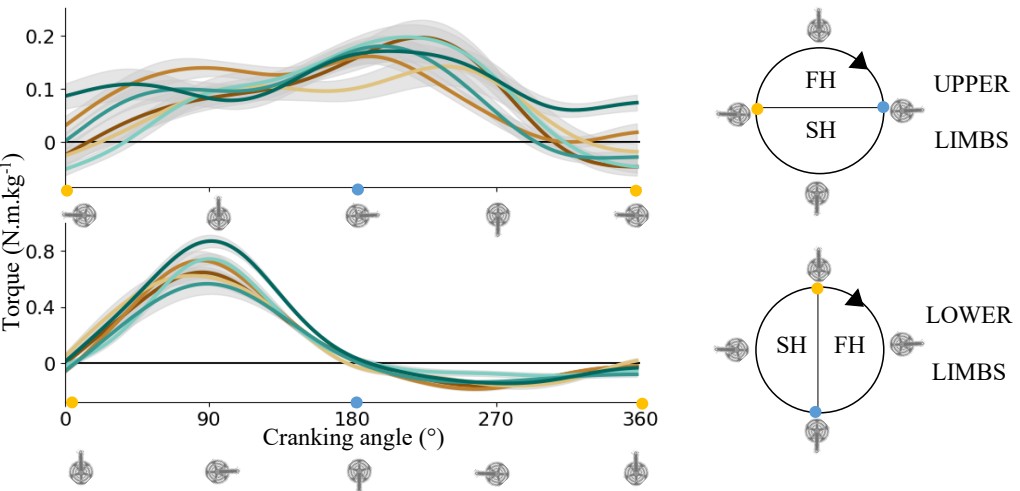

**Figure 3 Crank torque across the cycling revolution.** Mean ± sd right crank torque of each subject for upper limbs (top) and lower limbs (bottom). A schematic representation of the cycling revolution. The blue point illustrates the FH to SH transition, and the yellow point illustrates the SH to FH transition or upper transition for upper and lower limbs respectively. FH, First Half; SH, Second Half.

$p < 0.05$, excepted for TA and TF, Table 2-EMG). See Figs. S3, and S4 for mean ± sd representation of each individual muscle pattern for UL and LL respectively.

### Synergies

The interindividual variability observed at the EMG level is reflected in the synergy activation coefficients, with greater interindividual variability for UL than LL ($VR_{Hs-UL}$ = 0.59 ± 0.13 > $VR_{Hs-LL}$ = 0,26 ± 0.18, $r_{Hs-UL}$ = 0.38 ± 0.07 < $r_{Hs-LL}$ = 0.74 ± 0.17, $p < 0.05$). A larger time shift was found for all UL synergies ($t_{max-Hs-UL}$ = 12.96 ± 1.17 > $t_{max-Hs-LL}$ = 5.31 ± 3.73%, $p < 0.05$) despite a good index of similarity at the maximum of $r_{max}$ function ($r_{max-Hs-UL}$: = 0.87 ± 0.05 and $r_{max-Hs-LL}$ = 0.93 ± 0.06). UL and LL $W_s$ showed similar correlation coefficients (0.74 ± 0.02 vs 0.69 ± 0.21 for UL and LL respectively) excepted for $S_3$ of LL ($r_{S3}$ = 0.39 ± 0.37).

### Correlation in time shifts

The correlations in time shifting of the synergies were all significant for UL (Fig. S1, $r_{UL}$ > 0.81, $p_{UL}$< 0.00) and only for $S_2$ and $S_3$ on LL (Fig. S2, $r$ = 0.70, $p < 0.00$). This result indicates that all synergy activation coefficients are time-shifted by a similar amount for each participant and thus, despite the high interindividual variability in EMG patterns, participants produced the same coordination pattern (*i.e.*, 4 similar synergies, produced at similar relative interval) but with delayed timings.

A weak negative correlation is observed between all UL synergy activation coefficients and RPM ($r_{RPM-H1}$ = −0.58, $r_{RPM-H2}$ = −0.50, $r_{RPM-H3}$ = −0.50, $r_{RPM-H4}$ = −0.49, $p$ < 0.00) and to a lesser extent for LL synergy activation coefficients and RPM ($r_{RPM-H1}$ = −0.28, $r_{RPM-H2}$ = −0.35, $r_{RPM-H4}$ = −0.55, $p < 0.00$). No correlation was found between $H_s$ and torque for UL.

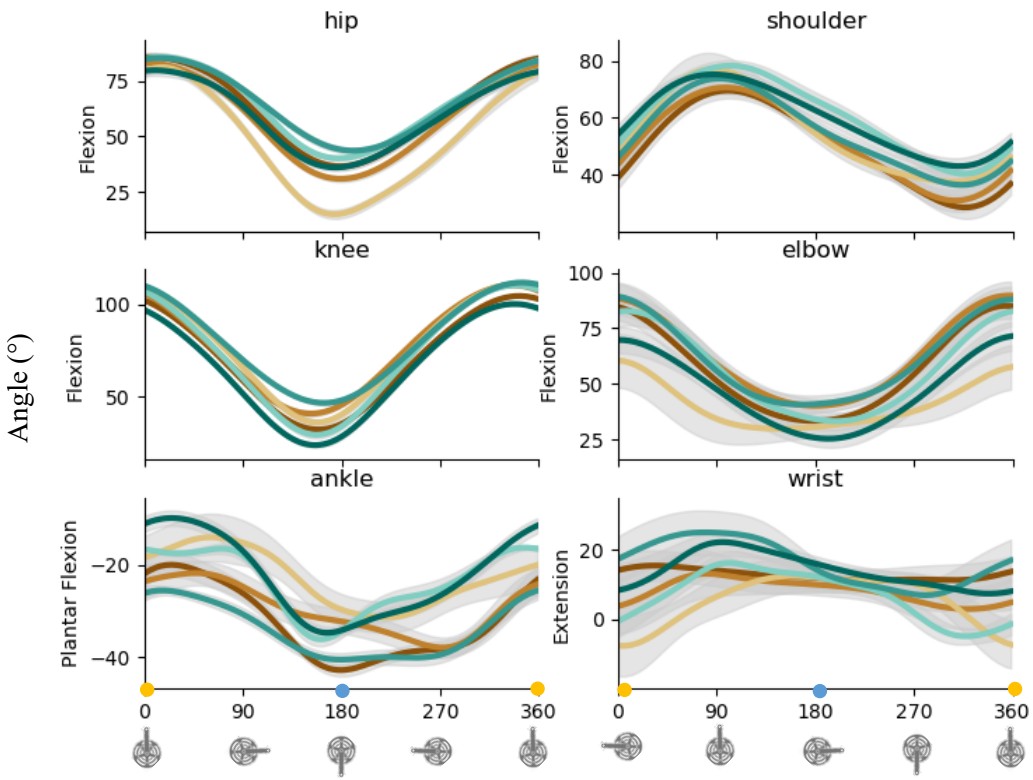

**Figure 4 Kinematic pattern across the cycling revolution.** Mean ± sd angle (degree) of proximal, intermediate, and distal articulations of LL (left) and UL (right) torque of each subject for UL (top) and LL (bottom). The blue point illustrates the FH to SH transition, and the yellow point illustrates the SH to FH transition or upper transition for upper and lower limbs respectively. FH, First Half; SH, Second Half; UL, upper limbs; LL, lower limbs.

### Intraindividual variability

The results concerning the interindividual variability are described in Tables 3 and 4. Intraindividual variability was inferior to interindividual variability and showed a good repeatability of kinematics, torque, EMG, and synergy activation coefficients for each participant.

## DISCUSSION

The first objective of this study was to investigate whether common muscular coordination processes related to mechanical output were observable during upper limbs (UL) and lower limbs (LL) cycling. To this aim, muscles coordination was analyzed through synergies in relation to the torque produced during cycling for both limbs. Even though the movement is strongly constrained, and the relative task intensity is similar for UL and LL, the torque outputs produced by both limbs showed strong differences. The results showed that cycling with the LL resulted in a reciprocal contribution of one leg then the other and a production of propulsive torque during half the cycle only. Cycling with the UL resulted in a different torque pattern arising from a simultaneous contribution of both arms, each producing

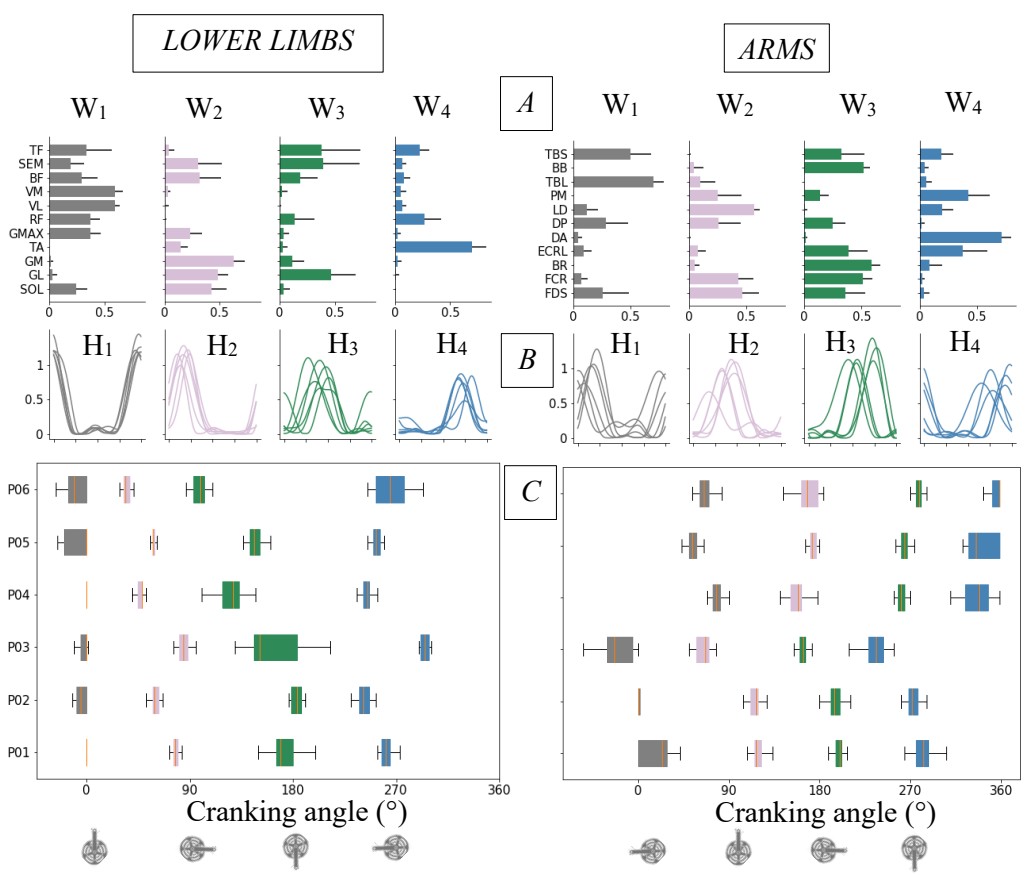

**Figure 5 Muscle's synergy parameters and variability.** Muscle synergies for LL (left) and UL (right). (A) Muscle's synergies vectors ($W_s$) for each muscle (mean ± sd). (B) Mean synergy activation coefficient ($H_s$) for each subject. (C) Boxplot of activation coefficient peak occurrences across the cycle revolution during 25 cycles for each subject (each row). The boxplot showed interindividual variability in the occurrence of the peaks for each activation coefficient: the participants 1 and 2 showed synchronous synergy occurrence, the participant 3 showed an early peak for each synergy and the participants 4, 5 and 6 showed latter peaks for each synergy. The detail of $r_{\text{inter-H}}$ showed high correlation between participants 1 and 2 ($r_{\text{inter-1/2}} = 0.95$) and between participants 4, 5 and 6 ($r_{\text{inter-4/5/6}} > 0.94$). The third participant showed weak or negative correlation with all the others.

propulsive torque during most of the cycle. Despite these differences in mechanical output, the same number of muscle synergies (4) were extracted to account for the variations in muscle activations of 11 muscles for each limb. The simultaneous analysis of the synergies related to the mechanical output showed different functional implications in the movement production between UL and LL.

Our second objective was to compare the interindividual and intraindividual variability of the UL and LL torque, joint kinematics, EMG, and synergy-related variables. The results showed a higher variability of the UL in kinematics, torque, EMG and in muscle coordination compared to LL. More precisely, the variability in EMG patterns was mainly reflected in the temporal components of muscle synergies: each synergy activation coefficient ($H_s$) of UL showed important time shifts between participants. Nevertheless,
despite this significant variability, each participant, when analyzed individually, produced reproductible synergy activation coefficients.

Four synergies were extracted for both UL and LL, despite a large interindividual variability in EMG patterns for UL. Our results on muscle synergy vectors showed close values of correlation between participants for both limbs. The $r$-value obtained are lower than those obtained by *De Marchis et al. (2013)* or *Hug et al. (2010)*, which can certainly be explained by differences in the methodologies used (average EMG patterns vs 25 consecutive cycles) or by the type of population chosen (expert vs non-expert). It is thus reasonable to consider that the spatial constitution of the synergies is shared between individuals. This indicates that the structure of synergies was robust between participants and support the assumption of a spatially fixed synergies.

Concerning the temporal aspects of the synergies, our results showed a strong interindividual variability in the EMG patterns which is reflected in the activation coefficients of the synergies, as constated by several authors (*Torres-Oviedo & Ting, 2007*; *Safavynia & Ting, 2012*; *De Marchis et al., 2013*). More specifically, each UL temporal activation coefficients showed large interindividual variability, mainly resulting from differences in the onset in apparition of each synergy pattern between participants rather than differences in patterns produced (Table 1). These results showed that same four synergy activation patterns were produced between participants, with a constant interval between each synergy (Fig. S1), but the expression of these four synergies was highly variable in the timing of apparition during the cycle. In comparison, LL synergies, particularly Syn1 and Syn2 showed low inter-individual variability which is an important difference between UL and LL.

The four synergies extracted during LL cycling agree with the synergies described in the literature in novice participants (*De Marchis et al., 2013*) or professional cyclists (*Hug et al., 2010*). $S_1$ and $S_2$ were associated with the production of propulsive torque during the power phase (from top dead center to bottom dead center). These two synergies strongly mobilized the hip and knee extensor muscles and were therefore both involved in the same mechanical function: producing propulsive torque by pushing the pedal with a leg extension (Fig. 4 - left). $S_3$ and $S_4$ were expressed during the resistive torque production phase and may be involved to optimize the lower transition and minimize the resistive torque induced by the legs weight during the flexion of the limb (Fig. 4 –left). Importantly, a merging of $S_2$ and $S_3$ is frequently observed in literature with expertise. $S_3$ is associated with tangential application of the force on the pedal at the end of the push. A merging of theses synergies is currently depicted as better orientation of the force on the crank/pedal at the end of the propulsive phase. It is therefore not surprising that both showed variability and correlation in timing of apparition. Being related, if $S_2$ appears earlier, $S_3$ logically also appears earlier. The same explanation stands for the variability observed at the level of synergy vector ($W_3$).

To our knowledge, a single study looked at UL muscle synergies (*Botzheim et al., 2021*) and observed two distinct synergies and attributed theses synergies to the two phases in the movement: pushing and pulling. If that was enough to describe bimanual coordination, it is impossible to rely on this observation to describe the mechanical output. Indeed,

these authors did not used crank torque measurement nor indicates for the output power. Thanks to our methodology our study distinguishes a total of 4 synergies, three ($S_1$ to $S_3$) corresponded to propulsive torque and one ($S_4$) to the reduction of resistive torques. $S_1$ primarily involved the shoulder flexors and elbow extensor muscles meaning that the first increase in torque is related to a pushing action on the pedals. This synergy is simultaneous to shoulder antepulsion and elbow extension (Fig. 4 –right). $S_3$ involved the elbow flexors, the wrist/finger flexors, and the shoulder extensor. This synergy is simultaneous with the peak of propulsive torque production and indicates that pulling is primarily used to produce torque. This synergy is simultaneous to reduction of shoulder antepulsion and elbow flexion (*i.e.*, reducing limb length). Our results showed that a third synergy ($S_2$) was involved in the production (or conservation) of propulsive torque during the transition from FH to SH. This synergy involved the wrist/finger flexors and the *latissimus dorsi*, the muscle responsible for the lowering of the arm. It seems that this synergy makes it possible to produced/conserved propulsive torque by lowering the arm in the transition from FH to SH. The simultaneous reduction of shoulder antepulsion, while elbow is still extending is in line with this explanation. Furthermore, this synergy probably takes advantage of the gravitational force to optimize the lowering action by using the weight of the arm. However, given the high relative intensity of the cycling movement, the gravitational influence is probably minimal. As for the synergies observed on LL, the synergy $S_4$ may be responsible for minimizing the resistive torque around the transition from SH to FH.

These results demonstrated that for a similar task demand (*i.e.*, moving the crank at an identical relative intensity), the muscular coordination of the UL is organized around several mechanical functions (*i.e.*, pushing, lowering, and pulling) to produce propulsive torque while the LL produce propulsive torque using a single main function (*i.e.*, push). These differences lead to a higher mechanical efficiency for the UL pedaling (*i.e.*, lower maximal torque and longer propulsive phase). Such differences may result from the daily use of UL in tasks of reaching, gripping, and manipulating in 3D space. The musculoskeletal system could be optimized to both extend and flex the arm. The LL are used mainly in locomotion tasks thanks to a musculoskeletal system optimized for antigravity functions (therefore pushing upward and propelling the upper body). Consequently, with torque production over a greater proportion of the pedaling cycle, which is synonymous with greater mechanical efficiency in performing the movement, UL demonstrates an important adaptation capacity to perform this movement for which our subjects and most of the population are non -expert.

A main difference between UL and LL is the level of variability observed. The results showed a higher variability for the UL, both for torque (*performance* variability) and for EMG and muscles coordination (*realization* variability). Nevertheless, when considering the variability for the UL, our results showed less variability in *performance* in comparison with *realization*. Thus, a similar torque pattern is produced between participants, with minor differences in shape and amplitude. On the contrary, our results showed that the variability of *realization* is large and lies specifically in the temporal aspects. This observation is surprising since one would have expected that differences in the timing onsets of the synergies would imply similar differences in the timing onsets in torques production.

**Table 5** Cadence of each participant for UL (right) and LL (right).

|  | UL (RPM, mean ± sd) | LL (RPM, mean ± sd) |
|---|---|---|
| P01 | 90.6 ± 0.6 | 82.5 ± 0.3 |
| P02 | 78.1 ± 1.3 | 89.6 ± 2.0 |
| P03 | 99.6 ± 1.3 | 64.1 ± 0.7 |
| P04 | 71.4 ± 0.5 | 82.1 ± 1.8 |
| P05 | 92.6 ± 0.9 | 102.0 ± 0.8 |
| P06 | 81.7 ± 0.9 | 80.0 ± 1.1 |

**Notes.**

LL showed more interindividual differences in cadence (ranging from 64.1 ± 0.7 to 102.0 ± 0.8) than UL (ranging from 71.4 ± 0.5 to 99.6 ± 1.3). UL: upper limbs, LL, lower limbs.

This suggests that there was no direct link between the variability observed in muscle coordination and the variability of the resulting torque.

Several hypotheses can highlight these discrepancies: first, the difference in timing can be explained by differences in self-selected cadence as shown by the correlation results between the appearance of synergies and cadence. Nevertheless, the differences in cadence cannot totally explain the temporal variability of the synergy activation patterns, especially since on LL the results showed a wider range of cadence between participants and no phase shift in synergies (Table 5). Second, greater level of variability in the UL can also be explained by differences in expertise between the UL and LL cycling. Although the participants were chosen novices for both tasks, LL cycling is associated to greater expertise since it has been shown that cycling shares common neural circuits with walking (*Barroso et al., 2014*), which ultimately induces an expertise in the realization of cycling since all the participants knew how to walk. However, in the literature, it has been shown during a bench press exercise that a greater inter-individual variability in synergy activation patterns is synonym of expertise instead of non-experts (*Kristiansen et al., 2015*). Thus, a difference in expertise does not seem sufficient to explain the differences in realization variability observed between UL and LL.

Before that conclusion can be made on the topic of expertise, further studies are needed to understand whether expertise (multiple sessions of UL cycling) tends to reduce the realization variability between individuals or whether individual strategies persist. Nevertheless, from our results for "non-experts" (as most patients and athletes), a flexibility in the use of synergies is adopted when performing the movement with UL unlike LL for which robust coordination is adopted by everyone.

## CONCLUSION

This study compared the muscles coordination during cycling with Upper Limbs (UL) and Lower Limbs (LL). Torque profiles differed between UL and LL and UL EMG patterns showed a larger interindividual variability than LL. Nevertheless, as already reported for LL, a functional organization of the UL muscle activations do exist during UL cycling and is shared among participants. UL cycling is achieved with a higher flexibly in the temporal onsets of the synergies compared to LL cycling besides this higher flexibility in

realization was not reflected in the mechanical output. The overall results of this study could be considered when combining UL and LL cycling for an assessment/rehabilitation process. In particular for UL, our results showed that an individualized approach should be recommended to evaluate the coordination state of the patient and to adapt the exercise parameter.

### Funding
The authors received no funding for this work.

### Competing Interests
The authors declare there are no competing interests.

### Author Contributions
- Théo Cartier conceived and designed the experiments, performed the experiments, analyzed the data, prepared figures and/or tables, authored or reviewed drafts of the paper, and approved the final draft.
- Laurent Vigouroux, Elke Viehweger and Guillaume Rao conceived and designed the experiments, authored or reviewed drafts of the paper, and approved the final draft.

### Human Ethics
The following information was supplied relating to ethical approvals (*i.e.*, approving body and any reference numbers):
CERSTAPS approved the study (IRB00012476-2020-30-11-75).

### Data Availability
The data is available at FigShare: Cartier, Théo (2022): Data_2_Share.zip. figshare. Dataset. https://doi.org/10.6084/m9.figshare.19099754.v1.

### Supplemental Information
Supplemental information for this article can be found online at http://dx.doi.org/10.7717/peerj.13155#supplemental-information.

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
