# Peer review of "Subject specific muscle synergies and mechanical output during cycling with arms or legs"

_PeerJ, doi:10.7717/peerj.13155_

## Round 0.1 · original submission · Major Revisions

Whilst there are a number of minor corrections the reviewers suggest there are also some points that require more substantial consideration. Both reviewers especially noted the potential for the variability in the upper limb results affecting the conclusions and both ask that some explicit reporting of this be provided. Reviewer #1 also recommended addtional analyses be undertaken. If you are happy to undertake this work I would be willing to consider a revised manuscript.

Reviewer 1 ·

Basic reporting

English could be improved.

Experimental design

The experiment is original and the study was well conducted but the design is not well suited for the first research question (#1)

Validity of the findings

Some claims are not supported by any analysis.

Additional comments

Review of the paper entitled: Subject specific muscle synergies and mechanical
output during arms and legs cycling

The authors investigated the intra and inter subject variabilities in the structure of muscles coordination and in their mechanical output during leg and arms cycling. They found higher inter-individual variability in the temporal structure of the upper limbs muscles than in the lower limbs despite similar variability in the torque profiles.
The study is interesting and original but some claims are not supported by any analysis.
Major
-An important conclusion of the study is that for the arms the temporal recruitment of the synergies is highly flexible without strong repercussion in the mechanical output. However, the variability in mechanical output seems greater for the arms than for the legs (it should be tested) and it is not clear whether the relation between the variability in the synergy activations and the variability in the mechanical output is different or not between the two tasks.
-Torque represents the forces applied orthogonal to the crank, not all the forces, and the movement might be much more constrained (in the sagittal plane) for the lower limbs. For the lower limbs whole body stability might be greater too. The variability in muscle activations might simply reflect these two aspects.
-The functional interpretation of the synergies is likely biased by the impact of gravity on the different segments in the two tasks. However, this aspect is probably not the most important and as the authors commented “it is impossible to attribute a precise mechanical action for these synergies” (L330).
-The author should provide evidences that the differences in the temporal structure is not due to the differences in self-selected cadences, i.e., can you ascertain that small changes in the arm cycling cadence is not associated with large modifications of the muscles activation timing.
-I recommend using statistical tools to confirm some of the claims/conclusions. For example, the spatial structures (W) appear similar across participants while it was not the case for the temporal structure (L264-265). However, VR and Pearson’s r were computed using vectors of different dimensions. It is easier to obtain great correlations when comparing small vectors, therefore the differences might not be due to different “similarities” but to different vectors size. Using the same thresholds (moderate, strong) might not be very relevant here. Intra and inter similarities could be compared more formally also. Importantly, the analysis is dependent on the number of synergies extracted for each participant. Different approaches were used in the literature that do not depend on this number (e.g., Yang et al. 2019 or Frère and Hug, 2012).
Frère, J., & Hug, F. (2012). Between-subject variability of muscle synergies during a complex motor skill. Frontiers in computational neuroscience, 6, 99.

Yang, Q., Logan, D., & Giszter, S. F. (2019). Motor primitives are determined in early development and are then robustly conserved into adulthood. Proceedings of the National Academy of Sciences, 116(24), 12025-12034.

minor
Although the reviewer is not a native English speaker, the English sounds weird at different places and could probably be improved.
Despite the exclusion criterion (L 101), it is very likely that people are more used to the leg ergometer (a bike) than the arm ergometer.
L 88. The study is not really designed to investigate the first objective. The link between muscle coordination and mechanical output was simply inferred from the results but their relation was not really evaluated.
Were the feet attached to the pedal for the lower limbs?
L 171. Was this algorithm based on the kinematic data? Is so, which one?
L 157. Was the lag corrected (zero lag filter)?
L 232. Torque and body weight do not have the same dimensions. It is not exact to speak of percentages.
L 265. What does it mean that “P03 and P06 doesn’t showed the synergy S2” if four synergies were extracted for all of them and that “functional organization ... is shared among participants”. 2/6 of the participants represents a lot. Idem in L 282.
L 347. This interpretation is questionable: for the legs two synergies appear to push, one to make the transition (at the bottom dead center) and the last one seems to lift the leg (= pushing, lowering and pulling, respectively, for the arms).
L 364. The variability in Hug et al .2010 was related to the spatial structure.
L 373. This absence of correlation should be demonstrated.
The discussion should start with a resume of the main findings.

Reviewer 2 ·

Basic reporting

no comment

Experimental design

no comment

Validity of the findings

I have some concern regarding the interpretation of the results, as the analysis of the kinematics has not been shown and it could play an important role to explain the obtained results.

Additional comments

In this study, T. Cartier and co-workers analyze the muscle coordination underlying the execution of a cycling task with arms or legs. By using the muscle synergy framework, they show that the complexity of the coordination of 11 upper or lower limb muscles is the same and consists of the activation of 4 muscle synergies. By recording the crank torque, they also provide a biomechanical interpretation of the role of the identified muscle synergies. They conclude that arm cycling has a higher complexity in terms of biomechanical functions, as identified by the higher variability in synergy structure and recruitment.

The study faces an important topic related to motor control, with implications in the field of neurorehabilitation. Of particular relevance is the relationship between the neural circuits regulating the control of cyclical movements of arms and legs. The work is well introduced and properly contextualized. The method section is detailed and well described for repeatability. The results are also well presented. However, I have some doubts regarding few methodological points, as some definition should be clarified. Moreover, I have some concern regarding the interpretation of the results, as the analysis of the kinematics has not been shown and it could play an important role to explain the obtained results.

- The authors mention that reflective markers were used to calculate 2-D kinematics of upper and lower limb, but these results are not shown. Might the variability in the motor control strategies of the upper limb be related to a higher variability in kinematics? This is an important point to be clarified. The authors could show the kinematics tracks of the upper and lower limb, or even perform the VR analysis on these tracks. I would suggest at least to insert a figure showing the ankle, knee and hip on a panel and wrist, elbow and shoulder on another panel, similarly to what done in Figure 3.

- I have some doubts regarding the structure of the leg cycling muscle synergies. While S1, S2 and S4 have a functional significance/biomechanical interpretation in terms of co-active groups of muscles and are in line with previous literature, the structure of S3 seems quite strange. It is composed of the co-activation of GL and TF (I suppose it is Tensor Fasciae Latae, its acronym has not been defined at lines 145-148). How is it possible that GL works independently with respect to GM? And which is the biomechanical significance of such synergy?

- The definition of rinter and rintra is not clear, as it has not been explained in the text. Did you calculate all the possible pairs of r (intra or inter, escluding autocorrelation) and then use the average value? Please clarify this point.

- Line 63: activates – activated.

- Lines 218-220: the meaning of this sentence is not clear, I suggest rephrasing.

- Lines 223-227: I suggest rephrasing as the whole period sounds awkward. Also, in lines 223-225 it appears that there is no quantitative analysis, but only a visual analysis performed by the authors. Why not quantifying this information?

- Lines 232-233: what do you mean when expressing torque as a percentage of body weight?

- Equations 2 and 3, in the VAFtot and VAFm definition, a term (1 - ) is missing before the summation.

- I am not a native English speaker, but I think that English throughout the manuscript could be improved.

---

## Round 0.2 · accepted · Accept

Thank you for your comprehensive responses - the reviewers and I are fully satisfied with these. Congratulations on your manuscript acceptance.

Reviewer 2 ·

Basic reporting

no comment

Experimental design

no comment

Validity of the findings

no comment

Additional comments

I thank the authors for their additional analysis and for their extensive work of revision to the mansucript. All my previously raised concerns have been thoroughly addressed. I have no further comment related to the manuscript.